# Usefulness of Tumor Marker Score for Predicting the Prognosis of Hepatocellular Carcinoma Patients Treated with Atezolizumab Plus Bevacizumab: A Multicenter Retrospective Study

**DOI:** 10.3390/cancers15174348

**Published:** 2023-08-31

**Authors:** Kazunari Tanaka, Kunihiko Tsuji, Atsushi Hiraoka, Toshifumi Tada, Masashi Hirooka, Kazuya Kariyama, Joji Tani, Masanori Atsukawa, Koichi Takaguchi, Ei Itobayashi, Shinya Fukunishi, Toru Ishikawa, Kazuto Tajiri, Hironori Ochi, Hidenori Toyoda, Chikara Ogawa, Takashi Nishimura, Takeshi Hatanaka, Satoru Kakizaki, Noritomo Shimada, Kazuhito Kawata, Atsushi Naganuma, Hisashi Kosaka, Tomomitsu Matono, Hidekatsu Kuroda, Yutaka Yata, Hideko Ohama, Fujimasa Tada, Kazuhiro Nouso, Asahiro Morishita, Akemi Tsutsui, Takuya Nagano, Norio Itokawa, Tomomi Okubo, Taeang Arai, Keisuke Yokohama, Hiroki Nishikawa, Michitaka Imai, Yohei Koizumi, Shinichiro Nakamura, Hiroko Iijima, Masaki Kaibori, Yoichi Hiasa, Takashi Kumada

**Affiliations:** 1Center for Gastroenterology, Teine Keijinkai Hospital, Sapporo 006-8555, Japan; ktsuj@keijinkai.or.jp; 2Gastroenterology Center, Ehime Prefectural Central Hospital, Matsuyama 790-0024, Japan; hideko.ohama@ompu.ac.jp (H.O.); c-ftada@eph.pref.ehime.jp (F.T.); 3Department of Internal Medicine, Japanese Red Cross Himeji Hospital, Himeji 670-8540, Japan; tadat0627@ares.eonet.ne.jp (T.T.); s-nakamura@himeji.jrc.or.jp (S.N.); 4Department of Gastroenterology and Metabology, Ehime University Graduate School of Medicine, Toon 791-0295, Japan; masashih@m.ehime-u.ac.jp (M.H.); ykoizumi@m.ehime-u.ac.jp (Y.K.); hiasa@m.ehime-u.ac.jp (Y.H.); 5Department of Hepatology, Okayama City Hospital, Okayama 700-0962, Japan; kazuya_kariyama@okayama-gmc.or.jp (K.K.); nouso@cc.okayama-u.ac.jp (K.N.); 6Department of Gastroenterology and Hepatology, Kagawa University, Kagawa 761-0795, Japan; tani.joji@kagawa-u.ac.jp (J.T.); morishita.asahiro@kagawa-u.ac.jp (A.M.); 7Division of Gastroenterology and Hepatology, Department of Internal Medicine, Nippon Medical School, Tokyo 113-8603, Japanitokawa@nms.ac.jp (N.I.); ma6-0154@nms.ac.jp (T.O.); a7100058@nms.ac.jp (T.A.); 8Department of Hepatology, Kagawa Prefectural Central Hospital, Takamatsu 760-8557, Japan; k.takaguchi@chp-kagawa.jp (K.T.); a-tsutsui@chp-kagawa.jp (A.T.); t-nagano@chp-kagawa.jp (T.N.); 9Department of Gastroenterology, Asahi General Hospital, Asahi 289-2511, Japan; itobayas@crocus.ocn.ne.jp; 10Department of Gastroenterology, Osaka Medical and Pharmaceutical University, Osaka 569-8686, Japan; sh-fukunishi@hyo-med.ac.jp (S.F.); keisuke.yokohama@ompu.ac.jp (K.Y.); hiroki.nishikawa@ompu.ac.jp (H.N.); 11Department of Gastroenterology, Saiseikai Niigata Hospital, Niigata 950-1104, Japan; toruishi@ngt.saiseikai.or.jp (T.I.); michitaka2021cancer-center@niigata-cc.jp (M.I.); 12Department of Gastroenterology, Toyama University Hospital, Toyama 930-0194, Japan; tajikazu@med.u-toyama.ac.jp; 13Center for Liver-Biliary-Pancreatic Disease, Matsuyama Red Cross Hospital, Matsuyama 790-8524, Japan; hironori19810211@gmail.com; 14Department of Gastroenterology and Hepatology, Ogaki Municipal Hospital, Ogaki 503-8502, Japan; hmtoyoda@spice.ocn.ne.jp; 15Department of Gastroenterology, Japanese Red Cross Takamatsu Hospital, Takamatsu 760-0017, Japan; chikara.ogawa.19721202@gmail.com; 16Department of Gastroenterology and Hepatology, Hyogo Medical University, Nishinomiya 663-8501, Japan; tk-nishimura@hyo-med.ac.jp (T.N.); hiroko-i@hyo-med.ac.jp (H.I.); 17Department of Gastroenterology, Gunma Saiseikai Maebashi Hospital, Maebashi 371-0821, Japan; hatanaka@qk9.so-net.ne.jp; 18Department of Clinical Research, National Hospital Organization Takasaki General Medical Center, Takasaki 370-0829, Japan; kakizaki@gunma-u.ac.jp; 19Division of Gastroenterology and Hepatology, Otakanomori Hospital, Kashiwa 277-0863, Japan; noritomos@jcom.home.ne.jp; 20Hepatology Division, Department of Internal Medicine II, Hamamatsu University School of Medicine, Hamamatsu 565-0871, Japan; kawata@hama-med.ac.jp; 21Department of Gastroenterology, National Hospital Organization Takasaki General Medical Center, Takasaki 370-0829, Japan; naganuma.atsushi.nj@mail.hosp.go.jp; 22Department of Surgery, Kansai Medical University, Hirakata 573-1010, Japan; kosakahi@hirakata.kmu.ac.jp (H.K.); kaibori@hirakata.kmu.ac.jp (M.K.); 23Department of Hepatology, St. Mary’s Hospital, Himeji 670-0801, Japan; matomato@himemaria.or.jp; 24Division of Hepatology, Department of Internal Medicine, School of Medicine, Iwate Medical University, Morioka 020-8505, Japan; hikuro@iwate-med.ac.jp; 25Department of Gastroenterology, Hanwa Memorial Hospital, Osaka 558-0041, Japan; yu-yata@maebashi.saiseikai.or.jp; 26Department of Nursing, Gifu Kyoritsu University, Ogaki 503-8550, Japan; tkumada@gku.ac.jp

**Keywords:** hepatocellular carcinoma, tumor maker, atezolizumab plus bevacizumab, prognosis, predictive model

## Abstract

**Simple Summary:**

Atezolizumab plus bevacizumab (Atez/Bev) is the first-line treatment for unresectable advanced hepatocellular carcinoma (HCC). The tumor markers (TMs) for HCC include alpha-fetoprotein (AFP), fucosylated alpha-fetoprotein (AFP-L3), and des-gamma carboxyprothrombin (DCP). A TM score combining these markers has been reported to be useful in predicting HCC prognosis. This retrospective study aimed to evaluate the ability of this previously reported TM score involving AFP, AFP-L3, and DCP as TMs in predicting prognosis and therapeutic efficacy in HCC patients administered Atez/Bev as first-line treatment. The TM score was found to be effective in stratifying overall survival and progression-free survival in 371 patients with unresectable advanced HCC treated with Atez/Bev. The TM score proved to be a simple and useful prognostic marker and therapeutic efficacy indicator for advanced HCC patients administered Atez/Bev as first-line treatment.

**Abstract:**

Aim: This study aimed to evaluate the ability of a previously reported tumor marker (TM) score involving alpha-fetoprotein (AFP), fucosylated AFP (AFP-L3), and des gamma-carboxy prothrombin (DCP) as TMs in predicting the prognosis and therapeutic efficacy in hepatocellular carcinoma (HCC) patients administered atezolizumab plus bevacizumab (Atez/Bev) as first-line treatment. Materials/Methods: The study period covered September 2020 to December 2022 and involved 371 HCC patients treated with Atez/Bev. The values of the TMs AFP, AFP-L3, and DCP were measured upon introducing Atez/Bev. Elevations in the values of AFP (≥100 ng/mL), AFP-L3 (≥10%), and DCP (≥100 mAU/mL) were considered to indicate a positive TM. The number of positive TMs was summed up and used as the TM score, as previously proposed. Hepatic reserve function was assessed using the modified albumin–bilirubin grade (mALBI). Predictive values for prognosis were evaluated retrospectively. Results: A TM score of 0 was shown in 81 HCC patients (21.8%), 1 in 110 (29.6%), 2 in 112 (29.9%), and 3 in 68 (18.3%). The median overall survival (OS) times for TM scores 0, 1, 2, and 3 were not applicable [NA] (95% CI NA-NA), 24.0 months (95% CI 17.8-NA), 16.7 months (95% CI 17.8-NA), and NA (95% CI 8.3-NA), respectively (*p* < 0.001). The median progression-free survival (PFS) times for TM scores 0, 1, 2, and 3 were 16.5 months (95% CI 8.0-not applicable [NA]), 13.8 months (95% CI 10.6–21.3), 7.7 months (95% CI 5.3–8.9), and 5.8 months (95% CI 3.0–7.6), respectively (*p* < 0.001). OS was well stratified in mALBI 1/2a and mALBI 2a/2b. PFS was well stratified in mALBI 2a/2b, but not in mALBI 1/2a. Conclusions: The TM score involving AFP, AFP-L3, and DCP as TMs was useful in predicting the prognosis and therapeutic efficacy in terms of OS and PFS in HCC patients administered Atez/Bev as first-line treatment.

## 1. Introduction

Following the results of the IMbrave150 trial [1], the combination therapy of atezolizumab bevacizumab (Atez/Bev) has been identified as the most promising systemic treatment for advanced hepatocellular carcinoma (HCC). Atez is a programmed cell death ligand 1 (PD-L1) inhibitor, and Bev is an anti-vascular endothelial growth factor antibody. Although, the objective response rate (ORR) and disease control rate (DCR) of Atez/Bev are reportedly 30% and 74% [2], respectively, there has been a need to develop a sensitive prognostic stratification tool in HCC patients treated with Atez/Bev. Previously proposed staging systems for HCC include the Barcelona Clinical Liver Cancer (BCLC) staging [3], Cancer of the Liver Italian Program (CLIP) score [4], Japan Integrated Staging (JIS) [5], and Albumin–Bilirubin-TNM of the Liver Cancer Study Group of Japan 6th edition (LCSGJ 6th) (ALBI-T) score [6]. These systems consist of tumor burden and hepatic reserve function. On the other hand, systemic chemotherapies are performed in patients with good hepatic reserve function.

However, there are only a few reports on prognostic scores for HCC that reflect the malignant potential of the tumor and the tumor microenvironment (TME) [7]. Immune checkpoint inhibitor efficacy is influenced by TME; therefore, a prognostic score reflecting TME is needed to predict the prognosis of HCC patients and the efficacy of Atez/Bev. In Japan, three tumor markers (TMs), namely, alpha-fetoprotein (AFP), fucosylated AFP (AFP-L3), and des gamma-carboxy prothrombin (DCP), can be used not only for surveillance of HCC, but also for assessment of the malignant potential of HCC in Japan. These TMs have recently been reported to be associated with tumor growth [8,9] and TME [10,11,12] in HCC. Although several studies have evaluated the association between the number of positive TMs and the prognosis [13,14,15,16,17], to our knowledge, there have been no reports on such an association in HCC patients treated with Atez/Bev.

The present study aimed to evaluate the ability of a previously proposed TM score involving AFP, AFP-L3, and DCP as TMs [14,15] in predicting the prognosis and therapeutic efficacy in HCC patients administered Atez/Bev as first-line treatment.

## 2. Materials and Methods

### 2.1. Patients

From September 2020 to December 2022, 749 Japanese HCC patients were treated with Atez/Bev at our institutions in Japan. After the exclusion of patients with a systemic pharmacotherapy history or patients without clinical data regarding TMs, we analyzed the clinical characteristics and prognosis of 371 patients retrospectively (Figure 1).

The decision to introduce Atez/Bev was made by the attending physician at each of the participating institutions. Treatment with intravenous Atez/Bev, which is composed of 1200 mg of atezolizumab plus bevacizumab at 15 mg/kg of body weight, was given every three weeks [1] based on the guidelines provided by the manufacturer. Treatment was discontinued following observation of any unacceptable or serious adverse event (AE) or clinical tumor progression. Each patient underwent an upper-gastrointestinal endoscopy examination for the surveillance of esophago-gastric varices (EGVs) within six months of the introduction of Atez/Bev. When bleeding was detected or in cases with a high risk (EGV F2 or more, positive for red color sign), endoscopic treatment (variceal ligation, or injection sclerotherapy) was administered before starting Atez/Bev therapy.

After receiving official approval, this study was conducted as a retrospective analysis of database records, based on the Guidelines for Clinical Research issued by the Ministry of Health and Welfare of Japan. All procedures were conducted in accordance with the Declaration of Helsinki. Written informed consent was received from each of the enrolled patients.

### 2.2. Basal Hepatic Disease and Assessment Methods for Hepatic Function

Patients with a positive anti-hepatitis C virus (HCV) test result were judged to have HCC due to HCV, and patients with a positive hepatitis B virus (HBV) surface antigen (HBsAg) test result were judged to have HCC due to HBV. Patients with alcohol-induced HCC were defined as having a history of excessive alcohol consumption of >60 g/day for five years or more. Patients not matching the above features were classified as others. Child–Pugh classification [18], albumin–bilirubin (ALBI) grade [19], and modified ALBI (mALBI) grade, in which ALBI grade 2 was divided into two subgrades (2a and 2b) using an ALBI score of −2.27 as the cut-off value [20], were used to assess hepatic reserve function.

### 2.3. Diagnosis and Therapeutic Strategies for HCC

HCC was diagnosed based on hyperattenuation at the arterial phase or hypoattenuation at the portal phase, as determined using dynamic computed tomography (CT) or magnetic resonance imaging (MRI) with tumor staining on angiography. Tumor stage was evaluated using the BCLC stage. The treatment strategy for HCC in this study followed the *Clinical Practice Guidelines for Hepatocellular Carcinoma* published by the Japan Society of Hepatology [21].

### 2.4. Assessment of Treatment Response

Treatment response was assessed using dynamic CT or MRI, according to the Response Evaluation Criteria in Solid Tumors (RECIST ver. 1.1) [22]. The overall response rate (ORR) and disease control rate (DCR) were calculated based on the radiological response. ORR was defined as the complete response or partial response of patients. DCR was defined as the complete response, partial response, and stable disease of patients. Overall survival (OS) was defined as the time from the date of Atez/Bev treatment initiation to the date of death or last visit. Progression-free survival (PFS) was defined as the time from Atez/Bev treatment initiation to the observation of clinical disease progression or death.

### 2.5. Methods for Assessment of Prognosis

The TM score consisted of AFP, AFP-L3, and DCP, and was measured before the start of Atez/Bev therapy. Elevations in the AFP (≥100 ng/mL), AFP-L3 (≥10%), and DCP (≥100 mAU/mL) values were defined as 1 point each, according to previous reports [15]. A prognostic score was generated by summing up the number of positive TMs. All patients were graded according to the TM score from 0 to 3. TM score, BCLC stage, Japan Integrated Staging (JIS), and mALBI-T were used for the evaluation of the prognosis, and their predictive abilities for OS and PFS were compared retrospectively. JIS was calculated using the Child–Pugh class and TNM LCSGJ 6th edition and had 6 grades from scores 0 to 5. mALBI-T was similarly calculated using the mALBI grade, instead of the Child–Pugh class, and the TNM LCSGJ 6th edition and had 7 grades from scores 0 to 6.

### 2.6. Statistical Analysis

Median values and the interquartile range (IQR) were used to express continuous variables. Statistical analysis was performed using Student’s *t*-test, Fischer’s exact test, and Mann–Whitney’s U-test. After the introduction of Atez/Bev, PFS and OS were evaluated using the Kaplan–Meier method and log-rank test. Clinical prognostic factors for PFS and OS were analyzed using Cox hazard analysis with age, gender (female), etiology (viral), mALBI grade ≥ 2b, BCLC stage, and TM score. The concordance index (c-index) was used for the evaluation of the ability for stratification and prediction of prognosis by each method. The Holm’s method was used for multiple comparisons. A *p*-value of <0.05 was considered to indicate a statistically significant difference. All statistical analyses were performed using Easy R (EZR), ver. 1.56 (Saitama Medical Center, Jichi Medical University, Saitama, Japan) [23]. 

## 3. Results

### 3.1. Characteristics of the HCC Patients

The median age of the 371 HCC patients was 74 years, with 78.4% being men. About 80% of these patients showed an Eastern Cooperative Oncology Group performance status (ECOG PS) of 0, and their median body mass index (BMI) was 23.6 kg/m^2^ (IQR 21.2–26.1). The median observation period was 10.4 months. The basal liver diseases were viral hepatitis (*n* = 180, 48.5%), alcoholic liver disease (*n* = 76, 20.5%), non-alcoholic steatohepatitis (*n* = 29, 7.8%) and others (*n* = 86, 23.2%).

A Child–Pugh score of 5 was shown in 225 (60.6%) patients, a score of 6 in 103 (27.8%), and a score of 7 or more in 43 (11.6%). The median ALBI score was −2.47 (IQR −2.75–−2.14). mALBI grade 1 was shown in 144 (38.8%) patients, grade 2a in 93 (25.1%), grade 2b in 126 (34.0%), and grade 3 in 8 (2.2%). BCLC stage 0 was shown in 4 (1.1%) patients, stage A in 18 (4.9%), stage B in 142 (38.3%), stage C in 196 (52.8%), and stage D in 11 (3.0%) (Table 1).

### 3.2. Prognosis of HCC Patients Treated with Atez/Bev

For all 371 patients, the median OS was not applicable [NA] (95% CI 20.1-NA), and the median PFS was 8.9 months (95% CI 7.3–12.6) (Figure 2a,b). During the follow-up period, 106 patients (28.6%) died, and 187 (50.4%) had invalid discontinuation of Atez/Bev. The best radiological responses using RECIST ver. 1.1 were as follows: complete response (CR) in 17 (5.0%) patients, partial response (PR) in 87 (25.6%), stable disease (SD) in 172 (50.6%), and progressive disease (PD) in 64 (18.8%) (after the exclusion of patients without an imaging evaluation: *n* = 31). In the present cohort, there was no statistically significant difference in OS and PFS among different etiologies of underlying liver diseases (Figure 2c,d).

### 3.3. Tumor Marker Score

Among the 371 subjects, 130 (35.0%) showed positivity for AFP (≥100 ng/mL), 186 (50.1%) for AFP-L3 (≥10%), and 220 (59.3%) for DCP (≥100 mAU/mL). A TM score of 0 was shown in 81 (21.8%) HCC patients, a score of 1 in 110 (29.6%), a score of 2 in 112 (29.9%), and a score of 3 in 68 (18.3%) (Figure 3). Of the 110 with a TM score 1, 7 (6.4%) were AFP-positive.

### 3.4. Overall Survival and Progression-Free Survival According to Elevation of Each Tumor Marker

OS was better in patients without TM elevation than in patients with TM elevation, according to the TM: the median OS (mOS) with AFP (<100 ng/mL) was NA (95% CI: 24.0-NA), whereas the mOS with AFP (≥100 ng/mL) was 16.7 months (95% CI: 11.3-NA) (*p* < 0.001) (Figure 4a). The mOS with AFP-L3 (<10%) was NA (95% CI: 24.0-NA), whereas the mOS with AFP-L3 (≥10%) was 17.8 months (95% CI: 15.0-NA) (*p* < 0.001) (Figure 4b). The mOS with DCP (<100 mAU/mL) was NA (95% CI: 20.1-NA), whereas the mOS with DCP (≥100 mAU/mL) was 24.0 months (95% CI: 13.5-NA) (*p* < 0.001) (Figure 4c).

The approximate HRs of each TM for OS were as follows: AFP (≥100 ng/mL) (HR 1.890), AFP-L3 (≥10%) (HR 1.928), and DCP (≥100 mAU/mL) (HR 1.915) (Table 2).

The median PFS (mPFS) was 13.4 months (95% CI: 10.2–20.3) for AFP (<100 ng/mL) and 6.3 months (95% CI: 4.5–8.3) for AFP (≥100 ng/mL) (*p* < 0.001) (Figure 4d). The mPFS was 16.3 months (95% CI: 9.6-NA) for DCP (<100 mAU/mL) and 7.5 months (95% CI: 6.1–8.9) for DCP (≥100 mAU/mL) (Figure 4e). AFP-L3 did not show a significant difference in PFS between patients without and with elevation [11.8 months (95% CI: 8.0–16.3) vs. 7.6 months (95% CI: 6.3–8.9)] (Figure 4f). However, the HRs for PFS according to TM were nearly similar for AFP (≥100 ng/mL) and DCP (≥100 mAU/mL), but not for AFP-L3 (>10%) (HRs: AFP 1.865, DCP 1.636, and AFP-L3 1.230) (Table 2).

### 3.5. Predictive Value According to TM Score, BCLC Stage, JIS, and mALBI-T Score

The numbers of patients with a radiological response are shown in Table 3. The ORR/DCR was 31.9%/80.6% for TM score 0, 29.5%/88.6% for score 1, 35.3%/79.4% for score 2, and 23.0%/7.21% for score 3, with no significant differences between the groups.

The mOS times for TM scores 0, 1, 2, and 3 were NA (95% CI NA-NA), 24.0 months (95% CI 17.8-NA), 16.7 months (95% CI 12.3-NA), and NA (95% CI 8.3-NA), respectively (*p* < 0.001, c-index 0.658) (Figure 5a). The HR for death increased with increasing positive number of TMs (HR: TM score 1 = 2.525, TM score 2 = 3.796, TM score 3 = 4.977) (Table 2). Although all the integrated scores were able to stratify the survival of patients, the c-index for the TM score (0.658) was equal to that of the mALBI-T score (0.658) and superior to those of the BCLC stage (0.577) and JIS (0.586) in the present unresectable HCC patients treated with Atez/Bev.

The mPFS times for TM scores 0, 1, 2, and 3 were stratified as 16.5 months (95% CI 8.0-NA), 13.8 (95% CI 10.6–21.3), 7.7 months (95% CI 5.3–8.9), and 5.8 months (95% CI 3.0–7.6), respectively (*p* < 0.001; c-index 0.594) (Figure 5b). The HR for PFS showed no significant difference between patients with a TM score of 0 and a TM score of 1, but the HR increased as the number of scores became more than a TM score of 2 (HR: TM score 2 = 1.595, score 3 = 2.345) (Table 2). Although all the integrated scores were able to stratify the mPFS, the c-index of the TM score (0.594) was superior to those of mALBI-T (0.547), BCLC stage (0.580), and JIS (0.554).

When the abilities of the TM score to predict the prognosis of HCC patients in terms of OS and PFS were examined according to mALBI 1/2a vs. mALBI 2b/3, OS was favorably stratified in the patients with mALBI 1/2a by TM score (*p* = 0.023, c-index 0.63), as well as in the patients with mALBI 2b/3 (*p* = 0.004, c-index 0.66). PFS was well stratified in the patients with mALBI 2b/3 (*p* < 0.001, c-index 0.67), but not in the patients with mALBI 1/2a (*p* = 0.313, c-index 0.55) (Figure 6).

Examinations of the prognostic factors for death on Cox hazard univariate analysis showed that age (HR 1.035, 95% CI 1.011–1.060, *p* = 0.004), mALBI grade ≥ 2b (HR 2.804, 95% CI 1.908–4.120, *p* < 0.001), BCLC stage (HR 1.514, 95% CI 1.106–2.073, *p* = 0.009), and TM score (HR 1.586, 95% CI 1.307–1.924, *p* < 0.001) were significant prognostic factors. Furthermore, on multivariate analysis, TM score (HR 1.480, 95% CI 1.210–1.812, *p* < 0.001), mALBI grade ≥ 2b (HR 1.560, 95% CI 1.736–3.774, *p* < 0.001), and age (HR 1.037, 95% CI 1.013–1.061, *p* = 0.002) were also significant prognostic factors (Table 4).

## 4. Discussion

In the present study, we found two key points. First, the present TM score, which is calculated based on the positive number of TMs, showed a favorable prognostic stratification ability for OS and PFS in HCC patients who received Atez/Bev therapy as a first-line systemic chemotherapy. Second, although the stratification ability of the TM score for OS was comparable to that of the mALBI-T score, the TM score was superior to all the other integrated scores in predicting PFS.

Previously, Ryu et al. [13] described that the number of positive TMs correlates with the tumor size and prevalence of microvascular invasion, and is associated with a poorly differentiated pathological character of tumors. Kiriyama et al. [14] showed that the recurrence rate in HCC patients with a triple-positive TM was the highest among HCC patients after radical resection. Ueno et al. [17] reported that tumor recurrence after radiofrequency ablation is influenced by the number of positive TMs.

To the best of our knowledge, this is the first report to examine the prognostic stratification ability of the TM score in HCC patients treated with Atez/Bev. Several previous studies have shown that AFP, AFP-L3, and DCP can stratify the prognosis of HCC because AFP, AFP-L3, and DCP are considered to reflect the malignant potential of HCC [8,9,10,11,12]. An elevated AFP has been shown to have an association with vascular invasion, poor differentiation, and the presence of satellite lesions in HCC [24,25,26]. Moreover, AFP-L3 has been found to be associated with tumor burden (tumor number and tumor size) and the frequency of macrovascular invasion [27,28], whereas DCP has been found to be associated with tumor burden, occurrence of portal vein tumor invasion, and worse histological tumor grade [29,30,31,32]. Thus, it is thought reasonable to consider the TM score as a favorable prognostic predictive scoring system, even for HCC patients treated with Atez/Bev. The HRs for OS and PFS became larger with increasing TM score.

AFP and DCP have been reported to reflect not only tumor burden, but also TME. AFP is known to alter the proportion of CD4+/CD8+ T cells [33] and to tilt TME toward immunosuppression by suppressing dendritic cells [11] and NK cells [10]. DCP has also been reported to be associated with CTNNB1 mutations [34], and CTNNB1 mutations have been shown to be a factor for the poor response to the immune checkpoint inhibitor [12,35]. Therefore, the HR for PFS also increased with an increasing positive number of TMs. Namely, increasing TM score may indicate a worsening of the tumor microenvironment, potentially influencing the shortened duration of Atez/Bev efficacy. As a result, OS also may become shorter along with an increasing TM score. Thus, the present TM score has the advantage of being easily used and is applicable to daily clinical practice.

Although the total staging system for HCC, such as the CLIP score [3] and C-reactive protein and AFP in immunotherapy (CRAFITY) score [36], includes AFP, other scoring systems, such as BCLC staging [2], JIS [4], and the ALBI-T score [5], basically consist of the tumor burden and hepatic reserve function. As the selection of therapeutic modalities depends on the tumor burden and hepatic reserve function, these stagings are considered to be reasonable as a total staging system for predicting prognosis. On the other hand, systemic therapies for HCC including Atez/Bev are performed for unresectable HCC with a high tumor burden; however, these are recommended to be introduced in patients with better hepatic reserve function (Child–Pugh A). As a result, most of the patients who receive Atez/Bev therapy usually have a good hepatic function. Therefore, the present TM score, which reflects tumor grade and TME, is considered to have a better prognostic value than the conventional integrated score, which consists of tumor progression and liver reserve. While the CLIP score and CRAFITY score are prognostic scores that include AFP, AFP levels may not always increase. The TM score, which uses three tumor markers, is considered to have superior prognostic value compared to the conventional composite score based on tumor progression and liver reserve function in HCC patients treated with Atez/Bev treatment. In the near future, the development of prognostic scores specialized for Atez/Bev treatment, similar to the CRAFITY score based on the TM score, is expected.

In the present study, the predictive ability of the TM score for PFS was different between HCC patients with mALBI grades 1/2a and 2b/3. This might suggest that the hepatic reserve function may affect TME. In fact, liver cirrhosis is a state of immunodeficiency due to the excessive secretion of inflammatory cytokines [37], and a poorer hepatic reserve function, such as mALBI 2b/3, may worsen the microimmune environment and subsequently worsen TME, as implied by elevated TMs. OS was significantly longer than PFS in HCC patients with mALBI 1/2a; however, the difference between PFS and OS was not significant for each score in HCC patients with mALBI 2b/3. These results were thought to be due to the therapeutic efficacy of postprogression treatments. To obtain improvement of the prognosis in unresectable HCC, introducing systemic treatment is recommended in HCC patients with a better condition and better hepatic reserve function as early as possible. Although transarterial catheter chemo embolization (TACE) has been performed as the initial recommended treatment for BCLC-B patients, the concept of TACE-refractory for switching to systemic treatment before a decline in the hepatic reserve function cause by repeated TACE has been reported recently [38]. Previously, Hiraoka et al. [15] reported that TACE-refractory was more likely in BCLC-B patients with a TM score of 2 or greater. Moreover, Kudo et al. proposed the concept of TACE-unsuitable, indicating the clinical features of resistance to TACE [36]. The newest BCLC strategy [39] recommends the introduction of systemic treatment prior to TACE in TACE-unsuitable BCLC-B patients. Early transition from TACE to systemic pharmacotherapy or prior systemic pharmacotherapy can lead to a better prognosis. Based on the present results, wherein HCC patients with a TM score of <2 showed a very favorable prognosis when Atez/Bev was administered, Atez/Bev should be introduced to BCLC-B patients treated with TACE before reaching a TM score ≥ 2 if there is a poor response to treatment and TM values increase gradually. Recently, the therapeutic efficacy of immune checkpoint inhibitors (ICIs) has been suggested to be limited in HCC due to NASH [40]. In this study, similar to the findings reported by Hatanaka [41] and Espinoza [42], the prognosis of NASH-derived HCC treated with Atez/Bev was comparable to that of HCC due to a viral or alcohol etiology. These results suggest that in real-world clinical practice, the therapeutic efficacy of Atez/Bev for NASH-related HCC may not differ from that for other etiologies.

This study has several limitations. First, although this is a multicenter study with a large cohort size, this study has a retrospective nature. Second, the observation period may not be sufficient. Prospective studies with longer observation periods are recommended to obtain a more definitive conclusion. Third, although AFP ≥ 100 ng/mL, AFP-L3 ≥ 10%, and DCP ≥ 100 mAU/mL were used in this study based on previous reports, more optimal cut-off values for this prognostic predicting system should be examined for patients with unresectable HCC treated with Atez/Bev.

## 5. Conclusions

The TM score involving AFP, AFP-L3, and DCP as TMs was useful in predicting the prognosis and therapeutic efficacy in terms of OS and PFS in HCC patients administered Atez/Bev as a first-line treatment.

## Figures and Tables

**Figure 1 cancers-15-04348-f001:**
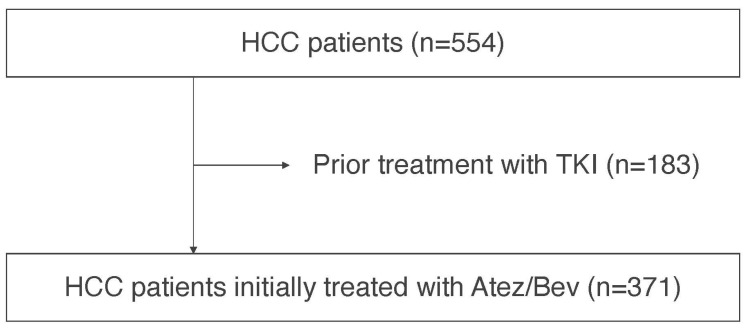
Patient flow diagram.

**Figure 2 cancers-15-04348-f002:**
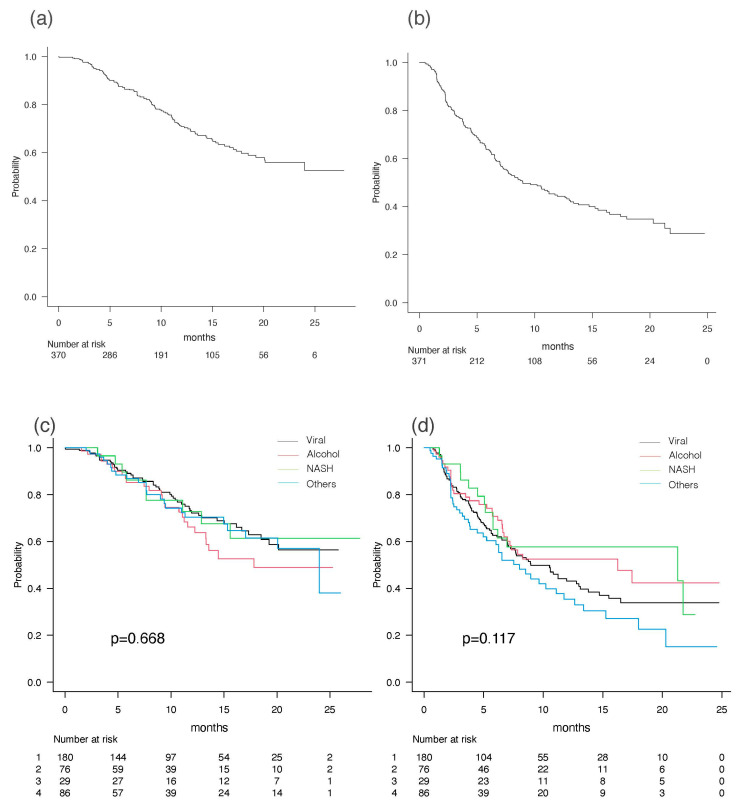
Overall and etiology-specific OS (**a**,**c**) and PFS (**b**,**d**).

**Figure 3 cancers-15-04348-f003:**
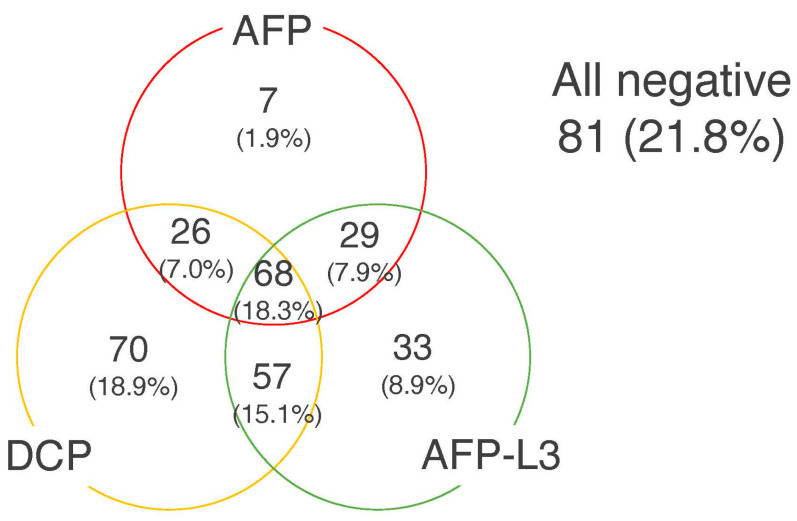
Distribution of patients with various patterns of positivity for the 3 tumor markers of HCC. The cut-off values of the tumor markers were as follows: AFP 100 ng/mL, AFP-L3 10%, and DCP 100 mAU/mL.

**Figure 4 cancers-15-04348-f004:**
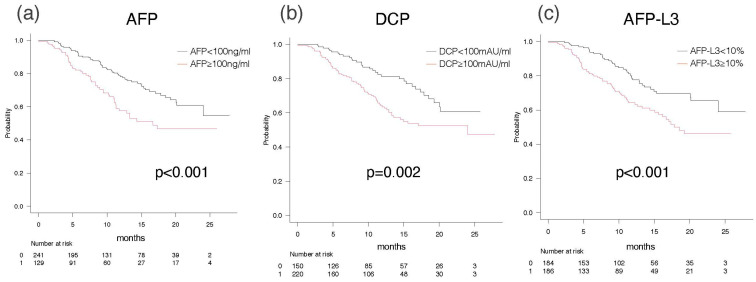
Overall survival and progression-free survival according to the elevations of each tumor marker. OSs according to the elevations of AFP (**a**), DCP (**b**), and AFP-L3 (**c**). PFSs according to the elevations of AFP (**d**), DCP (**e**), and AFP-L3 (**f**).

**Figure 5 cancers-15-04348-f005:**
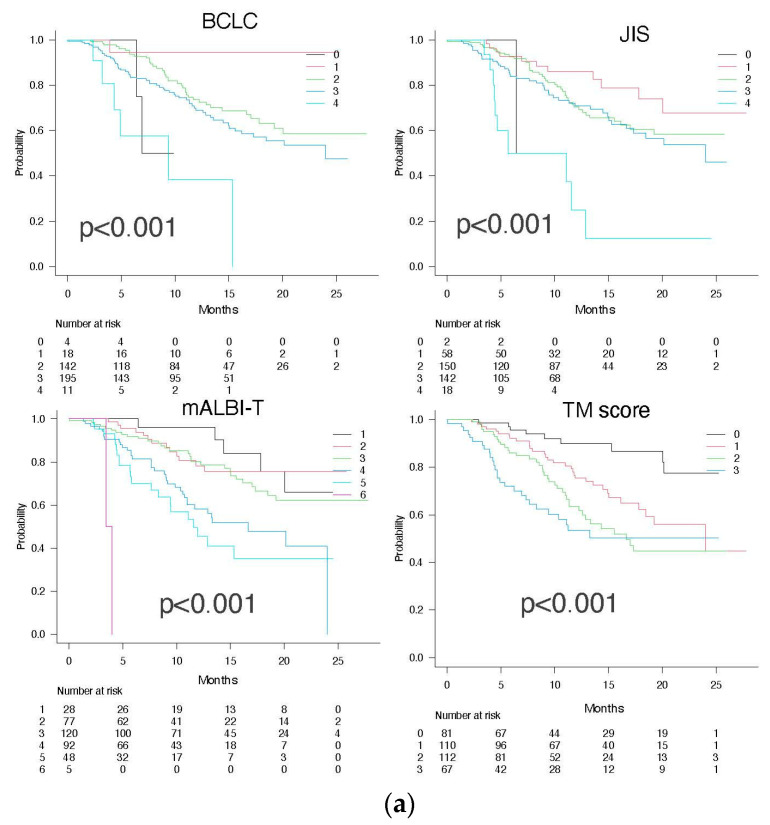
Overall survival (**a**) and progression-free survival (**b**) according to the BCLC stage, JIS, mATLBI-T score, and TM score.

**Figure 6 cancers-15-04348-f006:**
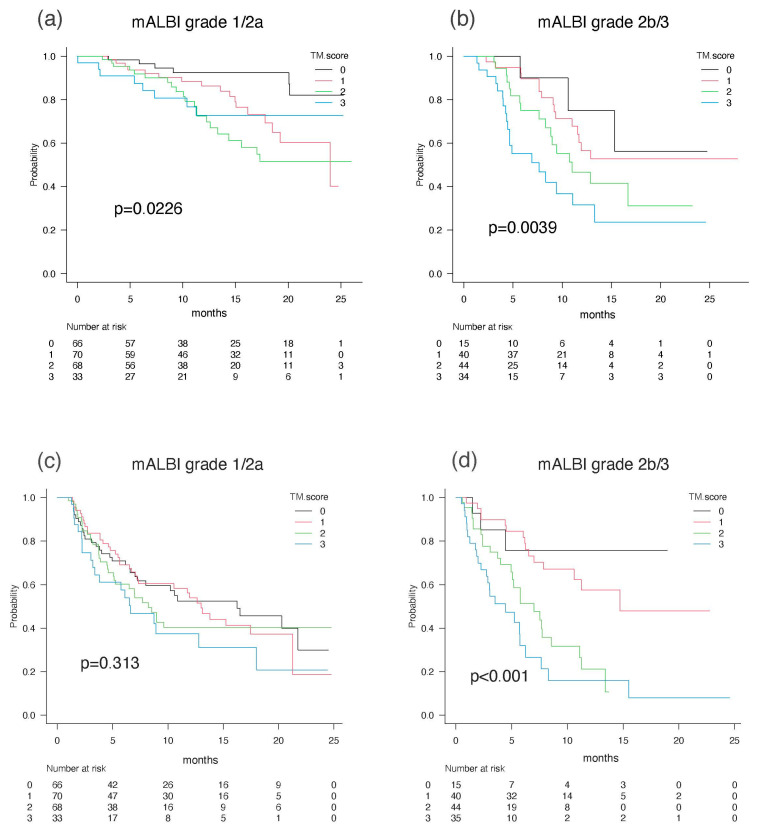
Overall survival and progression-free survival according to mALBI grade 1/2a and mALBI grade 2b/3. OSs for mALBI grade 1/2a (**a**) and 2b/3 (**b**). PFSs for mALBI grade 1/2a (**c**) and 2b/3 (**d**).

**Table 1 cancers-15-04348-t001:** Clinical Characteristics of the Study Patients.

Variables	Overall Patients	0 Points	1 Point	2 Points	3 Points	*p*-Value
Age, yrs *	74 [69, 81]	75 [70, 80]	74 [69, 79]	74 [69, 81]	75 [69, 81]	0.827
Gender, male: female	291:80	62:19	87:23	88:24	54:14	0.971
BMI, kg/m^2^ *	23.6 [21.2, 26.1]	24.5 [22.0, 26.5]	23.7 [21.1, 26.3]	23.2 [21.2, 26.1]	23.1 [20.6, 24.9]	0.170
ECOG PS						0.063
0	300 (80.9)	69 (85.2)	97 (88.2)	88 (78.6)	46 (67.6)	
1	55 (14.8)	10 (12.3)	12 (10.9)	17 (15.2)	16 (23.5)	
≧2	16 (4.3)	0	1 (0.9)	7 (6.3)	6 (8.9)	
Etiology of liver disease						0.320
HBV	128 (34.5)	31 (38.3)	36 (32.7)	39 (34.8)	22 (32.4)	
HCV	52 (14.0)	11 (13.6)	17 (15.5)	18 (16.1)	6 (8.8)	
Alcohol	76 (20.5)	14 (17.3)	27 (24.5)	15 (13.4)	20 (29.4)	
NASH	29 (7.8)	6 (7.4)	8 (7.3)	13 (11.6)	2 (2.9)	
Others	115 (31.0)	25 (30.9)	30 (27.3)	40 (35.7)	20 (29.4)	
BCLC stage						0.022
Very early (0)	4 (1.1)	1 (1.2)	0	2 (1.8)	1 (1.5)	
Early (A)	18 (4.9)	6 (7.4)	7 (6.4)	3 (2.7)	2 (2.9)	
Intermediate (B)	142 (38.3)	37 (45.7)	52 (47.3)	35 (31.2)	18 (26.5)	
Advanced (C)	196 (52.8)	35 (43.2)	50 (45.5)	69 (61.6)	42 (61.8)	
Terminal (D)	11 (3.0)	2 (2.5)	1 (0.9)	3 (2.7)	5 (7.4)	
Child–Pugh score, *n* (%)						0.003
5	225 (60.6)	52	77	65	32	
6	103 (27.8)	25	24	33	20	
≧7	43 (11.6)	4	9	14	16	
ALBI score *	−2.47 [−3.47, −0.81]	−2.60 [−2.95, −2.34]	−2.49 [−2.74, −2.20]	−2.40 [−2.68, −2.06]	−2.23 [−2.61, −1.92]	<0.001
mALBI grade, *n* (%)						0.006
1	144 (38.8)	41 (50.6)	45 (40.9)	39 (34.8)	19 (27.9)	
2a	93 (25.1)	25 (30.9)	25 (22.7)	29 (25.9)	14 (20.6)	
2b	126 (34.0)	15 (18.5)	39 (35.5)	39 (34.8)	33 (48.5)	
3	8 (2.2)	0	1 (0.9)	5 (4.5)	2 (2.9)	
Serum albumin (g/dL) *	3.8 [3.4, 4.1]	4.0 [3.6, 4.3]	3.8 [3.5, 4.1]	3.8 [3.4, 4.1]	3.6 [3.2, 3.9]	0.001
Total bilirubin (mg/dL) *	0.8 [0.6, 1.1]	0.7 [0.5, 1.0]	0.8 [0.6, 1.1]	0.8 [0.6, 1.1]	0.9 [0.7, 1.3]	0.002
Platelet count (10^9^/L) *	13.7 [10.2, 18.9]	13.4 [10.4, 18.7]	13.0 [10.5, 17.4]	13.8 [10.1, 19.5]	15.6 [10.3, 22.9]	0.216
Prothrombin time (%) *	88.0 [77.4, 99.0]	89.4 [84.9, 99.5]	91.0 [80.0, 100.0]	86.7 [77.0, 97.3]	83.5 [72.9, 97.0]	0.041
Extrahepatic spread, *n* (%)	110 (29.6%)	26 (32.1)	30 (27.3)	37 (33.0)	17 (25.0)	0.604
Macrovascular invasion, *n* (%)	59 (15.9)	3 (3.7)	11 (10.0)	22 (19.6)	23 (33.8)	<0.001
AFP ≧ 100 ng/mL, *n* (%)	130 (35.0)	0 (0)	7 (6.4)	55 (49.1)	68 (100)	<0.001
AFP-L3 ≧ 10%, *n* (%)	187 (50.4)	0 (0)	33 (30.0)	86 (76.8)	68 (100)	<0.001
DCP ≧ 100 mAU/mL	221 (59.6)	0 (0)	70 (63.6)	83 (74.1)	68 (100)	<0.001

* Median (interquartile range); BMI: body mass index, ECOG PS: eastern cooperative oncology group performance status, HBV: hepatitis B virus, HCV: hepatitis C virus, NASH: non-alcoholic steatohepatitis, BCLC: Barcelona clinical liver cancer, ALBI: albumin–bilirubin, mALBI: modified ALBI, AFP: alpha-fetoprotein, AFP-L3: fucosylated AFP, DCP: des-gamma-carboxy prothrombin.

**Table 2 cancers-15-04348-t002:** OS and PFS according to each tumor marker.

	OS			PFS		
	HR	95% CI	*p* Value	HR	95% CI	*p* Value
AFP ≥ 100 ng/mL	1.890	1.289–2.772	0.001	1.865	1.393–2.496	<0.001
AFP-L3 ≥ 10%	1.928	1.301–2.857	0.001	1.227	0.920–1.635	0.164
DCP ≥ 100 mAU/mL	1.915	1.264–2.901	0.002	1.636	1.208–2.215	0.001
Tumor marker score						
0	1.0			1.0		
1	2.525	1.205–5.294	0.014	1.022	0.661–1.581	0.923
2	3.796	1.834–7.857	<0.001	1.595	1.044–2.437	0.031
3	4.977	2.338–10.60	<0.001	2.345	1.495–3.677	<0.001

OS: overall survival, PFS: progression-free survival, AFP: alpha-fetoprotein, AFP-L3: fucosylated AFP, DCP: des-gamma-carboxy prothrombin.

**Table 3 cancers-15-04348-t003:** Confirmed radiological response rate according to tumor marker score.

	0 Points	1 Point	2 Points	3 Points	*p*-Value
Radiological response, *n* (%)					0.136
CR	5 (6.9)	7 (6.7)	3 (2.9)	2 (3.3)	
PR	18 (25.0)	24 (22.9)	33 (32.4)	12 (19.7)	
SD	35 (48.6)	62 (59.0)	45 (44.1)	30 (49.2)	
PD	14 (19.4)	12 (11.4)	21 (20.6)	17 (27.9)	
ORR, *n* (%)	23 (31.9)	31 (29.5)	36 (35.3)	14 (23.0)	0.416
DCR, *n* (%)	58 (80.6)	93 (88.6)	81 (79.4)	44 (72.1)	0.058

CR: complete response, PR: partial response, SD: stable disease, PD: progressive disease, ORR: objective response rate, DCR: disease control rate.

**Table 4 cancers-15-04348-t004:** Clinical prognostic factors.

	Univariate Analysis			Multivariate Analysis		
	HR	95% CI	*p* Value	HR	95% CI	*p* Value
Age	1.035	1.011–1.060	0.004	1.037	1.013–1.061	0.002
Gender (=Female)	0.772	0.469–1.268	0.307			
Etiology (=viral)	1.024	0.916–1.145	0.675			
mALBI grade ≥ 2b	2.804	1.908–4.120	<0.001	2.560	1.736–3.774	<0.001
BCLC stage	1.514	1.106–2.073	0.009	1.237	0.909–1.684	0.176
Tumor marker score	1.586	1.307–1.924	<0.001	1.480	1.210–1.812	<0.001

mALBI grade: modified albumin–bilirubin grade, BCLC: Barcelona clinical liver cancer.

## Data Availability

Due to the nature of this research, the participants could not be contacted as to whether the findings could be shared publicly; thus, supporting data, including datasets generated and/or analyzed for the current study, are not publicly available.

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
