# Peer review of "Usefulness of Tumor Marker Score for Predicting the Prognosis of Hepatocellular Carcinoma Patients Treated with Atezolizumab Plus Bevacizumab: A Multicenter Retrospective Study"

_cancers, 2023, doi:10.3390/cancers15174348_

Round 1

Reviewer 1 Report (Previous Reviewer 3)

The authors have addressed the previous concerns.

Reviewer 2 Report (Previous Reviewer 1)

Manuscript ID: cancers-2565998 - The manuscript “Usefulness of tumor marker score for predicting the prognosis of hepatocellular carcinoma patients treated with atezolizumab plus bevacizumab” by Kazunari Tanaka et al   has been satisfactorily revised. Authors have addressed all comments and now the manuscript warrants publication in” Cancers”.

Reviewer 3 Report (Previous Reviewer 2)

This manuscript has been appropriately revised.

This manuscript is a resubmission of an earlier submission. The following is a list of the peer review reports and author responses from that submission.

Round 1

Reviewer 1 Report

Report HCC (Cancers)

The manuscript “Usefulness of tumor marker score for predicting the prognosis of hepatocellular carcinoma patients treated with atezolizumab plus bevacizumab” by Kazunari Tanaka et al   is a retrospective study conducted with large cohort size.  This study highlights the prognostic significance of biomarkers, AFP, AFP-L3, and DCP in unresectable HCC patients treated with atezolizumab plus bevacizumab. Patients were dichotomized based on the median values either above the median or below (cut -off value) for these biomarkers and clinical outcome or survival benefit was estimated as OS or PFS using the Kaplan-Meir method and log-rank test. Several such studies have been reported in HCC patients treated with atezolizumab plus bevacizumab in the recent past, which limits the novelty the study.  Nevertheless, this study was sufficiently powered and well conducted with precision.  Please see below the reference you might want to incorporate in the discussion.

https://link.springer.com/article/10.1007/s12072-022-10358-z

Reviewer 2 Report

In this study, the authors investigated the ability of tumor marker score involving AFP, AFP-L3, and DCP in predicting prognosis and therapeutic efficacy in HCC patients administered Atez/Bev as first-line treatment.

This study has some limitations as the authors say, but I think it's a well-analyzed and interesting manuscript.

Check the following points.

1.      Can you show the P-value for each other than the TM in Figure 5?

2.      Is it necessary to underline 23.0%/7.21% on line 249?

3.      Please check the references number.

Reviewer 3 Report

The combination of atezolizumab plus bevacizumab (Atez/Bev) is now considered as the first line of therapy for HCC patients. In this manuscript the authors evaluated alpha-fetoprotein (AFP), fucosylated AFP (AFP-L3), and des gamma-carboxy pro-58 thrombin (DCP) as tumor markers (TM) to predict prognosis and therapeutic efficacy of Atez/Bev as first line treatment in 371 HCC patients in Japan. AFP (≥ 100 ng/ml), AFP-L3 63 (≥ 10%), and DCP (≥ 100 mAU/ml) levels, measured before the start of the treatment, were considered as positive TM which were used to make a composite TM score. It was shown than a higher TM score was associated with decreased median overall survival (OS) and progression-free survival (PFS) thus considered as useful for predicting prognosis and therapeutic efficacy.

This is a clinically relevant study which will contribute to better understanding of Atez/Bev treatment regimen and utilization of specific TMs. Overall the paper is well-written and the analysis is done comprehensively. There are some minor concerns that need to be addressed:

1.    HCC patients with underlying NASH have been reported to respond poorly to Atez/Bev treatment compared to those with viral hepatitis. It would be interesting to see how many of the patients who were classified as ‘others’ (i.e., not viral hepatitis or alcoholism) had underlying NASH and how they responded.

2.    It is also not clear if there is a differential response between patients with viral hepatitis and those with alcoholism.